# The *BRAF* P.V600E Mutation Status of Melanoma Lung Metastases Cannot Be Discriminated on Computed Tomography by LIDC Criteria nor Radiomics Using Machine Learning

**DOI:** 10.3390/jpm11040257

**Published:** 2021-04-01

**Authors:** Lindsay Angus, Martijn P. A. Starmans, Ana Rajicic, Arlette E. Odink, Mathilde Jalving, Wiro J. Niessen, Jacob J. Visser, Stefan Sleijfer, Stefan Klein, Astrid A. M. van der Veldt

**Affiliations:** 1Department of Medical Oncology, Erasmus MC Cancer Institute, 3015 GD Rotterdam, The Netherlands; a.rajicic@erasmusmc.nl (A.R.); s.sleijfer@erasmusmc.nl (S.S.); a.vanderveldt@erasmusmc.nl (A.A.M.v.d.V.); 2Department of Radiology and Nuclear Medicine, Erasmus MC, 3015 GD Rotterdam, The Netherlands; m.starmans@erasmusmc.nl (M.P.A.S.); a.odink@erasmusmc.nl (A.E.O.); w.niessen@erasmusmc.nl (W.J.N.); j.j.visser@erasmusmc.nl (J.J.V.); s.klein@erasmusmc.nl (S.K.); 3Department of Medical Informatics, Erasmus MC, 3015 GD Rotterdam, The Netherlands; 4Department of Medical Oncology, University Medical Center Groningen, University of Groningen, 9713 GZ Groningen, The Netherlands; m.jalving@umcg.nl; 5Faculty of Applied Sciences, Delft University of Technology, 2628 CJ Delft, The Netherlands

**Keywords:** melanoma, machine learning, lung neoplasm/metastases, tomography, X-ray computed, proto-oncogene proteins B-raf

## Abstract

Patients with *BRAF* mutated (*BRAF*-mt) metastatic melanoma benefit significantly from treatment with BRAF inhibitors. Currently, the *BRAF* status is determined on archival tumor tissue or on fresh tumor tissue from an invasive biopsy. The aim of this study was to evaluate whether radiomics can predict the *BRAF* status in a non-invasive manner. Patients with melanoma lung metastases, known *BRAF* status, and a pretreatment computed tomography scan were included. After semi-automatic annotation of the lung lesions (maximum two per patient), 540 radiomics features were extracted. A chest radiologist scored all segmented lung lesions according to the Lung Image Database Consortium (LIDC) criteria. Univariate analysis was performed to assess the predictive value of each feature for *BRAF* mutation status. A combination of various machine learning methods was used to develop *BRAF* decision models based on the radiomics features and LIDC criteria. A total of 169 lung lesions from 103 patients (51 *BRAF*-mt; 52 *BRAF* wild type) were included. There were no features with a significant discriminative value in the univariate analysis. Models based on radiomics features and LIDC criteria both performed as poorly as guessing. Hence, the *BRAF* mutation status in melanoma lung metastases cannot be predicted using radiomics features or visually scored LIDC criteria.

## 1. Introduction

Cutaneous melanoma is an aggressive skin cancer most commonly occurring on the ultra-violet light exposed skin of Caucasians [1,2]. In Europe, it is the 8th most common malignancy in men and the 5th most common in women, with an annual incidence of 144,200 new cases and 27,100 deaths [3]. In the coming years, the incidence of melanoma is expected to increase rapidly, resulting in an increased melanoma-associated mortality [4].

The introduction of new systemic treatment modalities, including immunotherapy and BRAF inhibitors, has significantly improved the prognosis of patients with metastatic melanoma [5]. Approximately 50% of melanomas harbor a mutation in the *BRAF* gene, with p.V600E being the most common variant [6,7,8]. Patients with *BRAF*-mutant (*BRAF*-mt) melanoma benefit significantly from treatment with BRAF inhibitors and onset of response is often rapid [9]. To enhance response rates and duration of response, patients are usually treated with a combination of a BRAF and a MEK inhibitor [10,11,12,13]. Due to the therapeutic consequences, determination of the *BRAF* mutation status in patients with metastatic melanoma is mandatory according to the European Society of Medical Oncology guidelines [14].

Currently, the *BRAF* mutation status is usually determined by molecular analysis of a metastatic lesion [15]. However, tissue biopsies are invasive, thereby exposing patients to potential risks including bleeding, infection and in case a lung biopsy is taken the risk of pneumothorax. In addition, molecular analyses can be time-consuming, especially when the tumor specimen has been archived at another hospital. Since patients with metastatic melanoma can experience rapidly progressive disease with life-threatening symptoms and an urgent medical need for systemic therapy, faster and less invasive diagnostics to determine the *BRAF* mutation status may significantly improve patient management.

Recently, various tumor characteristics have been predicted non-invasively using quantitative imaging features, also referred to as “radiomics”. In non-small cell lung cancer, radiomics on computed tomography (CT) can predict tumor stage and epidermal growth factor receptor (EGFR) mutation status [16,17,18,19,20,21,22,23,24]. In patients with primary colorectal cancer, a CT radiomics signature that was associated with *BRAF* mutation status [25]. CT-based radiomics has been applied to predict response to immunotherapy in melanoma lymph node metastases [26], but with little success (area under the curve (AUC) of 0.64). The value of radiomics for predicting *BRAF* mutation status has not been investigated. If CT-based radiomics could predict *BRAF* mutation status with a high positive predictive value, this may provide a faster and more patient-friendly alternative to determine the *BRAF* mutation status in metastatic melanoma.

The aim of this study was to evaluate the utility of CT-based radiomics to predict *BRAF* mutation status (mutant versus wild type) in metastatic melanoma. In metastatic melanoma, lung metastases are relatively easy to annotate on CT as compared to other metastases since they can be clearly distinguished from healthy lung tissue. Therefore, the aim of this study was to evaluate the utility of CT-based radiomics to predict *BRAF* mutation status (mutant versus wild type) in melanoma lung metastases.

## 2. Materials and Methods

### 2.1. Data Collection

This study was approved by the Erasmus MC institutional research board (MEC-2019-0693). Anonymized patient data was used and therefore need for written informed consent was waived by the Institutional Review Board. All patients diagnosed with metastatic melanoma at the Erasmus MC between January 2012 and February 2018 were included retrospectively if they met the following pre-specified criteria: known tumor *BRAF* mutation, diagnostic contrast-enhanced thoracic CT scan prior to commencement of any systemic therapy, and at least one lung metastasis of ≥10 mm evaluable according to Response Evaluation Criteria In Solid Tumors (RECIST) v1.1 [27]. Patients with *BRAF* mutations other than p.V600E were excluded from the analysis, since BRAF inhibitors may be less effective in patients with other *BRAF* mutations [28]. Formalin-fixed paraffin-embedded material of the primary tumor and/ or metastasis is tested for *BRAF* (exon 15) using a polymerase chain reaction based assay or next generation sequencing as part of standard care.

### 2.2. Radiomics

Lung metastases were measured according to RECIST v1.1 [27]. For 3D segmentation, up to two lung lesions ≥ 10 mm were selected by a clinician supervised by an experienced chest radiologist. In patients with >2 lung metastases of ≥10 mm, either the two largest or the two most easily distinguishable lesions were segmented (i.e., two separate lesions were preferred over two adjacent lesions). Using in-house developed software [29], selected lung metastases were segmented semi-automatically using a lung window for visualization. The result was visually inspected and manually corrected when necessary by an experienced chest radiologist to ensure that the semi-automatic segmentation resembled the manual segmentation. The clinician and chest radiologist were both blinded for *BRAF* mutation status. From each segmented lesion, 540 radiomics features were extracted to quantify intensity, shape and texture. Details are described in Appendix A. To create a decision model using these features, the Workflow for Optimal Radiomics Classification (WORC) toolbox was used (Figure 1) [30,31,32]. Details are described in Appendix A. In brief, the creation of a decision model in WORC consists of several steps, including selection of relevant features, resampling and machine learning techniques to identify patterns to distinguish *BRAF*-mt from *BRAF* wild type (*BRAF*-wt) lesions. WORC performs an automated search including a variety of algorithms for each step and determines which combination of algorithms maximizes the predictive performance on the training set. The open-source code for the feature extraction and model optimization has been published [33].

### 2.3. Scoring by Radiologist

An experienced chest radiologist (certified for 8 years) scored the segmented lung lesions. There are no guidelines to differentiate histologic subtypes in lung metastases; therefore, the Lung Image Database Consortium (LIDC) criteria were used. These criteria were developed to standardize the description of radiological features of lung abnormalities in clinical practice [34]. The following LIDC features were rated: subtlety, calcification, internal structure, lobulation, likelihood of malignancy, margin, sphericity, spiculation and texture (see Appendix A for the rating system). The radiologist was blinded for the *BRAF* status, but not to the diagnosis of metastatic melanoma and had access to the CT scan, age and sex of the patient.

### 2.4. Experimental Setup

To assess the predictive value of quantitative imaging features (i.e., radiomics features) and LIDC features, five models were trained and tested using WORC based on: (1) automatically extracted radiomics features only (2) similar to model 1, but only including the largest lesion per patient; (3) similar to model 1, but only including patients with *NRAS* and *BRAF* wild type melanoma for the comparison with *BRAF*-mt; (4) manually scored LIDC features only; and (5) a simple benchmark model. Model 2 was applied to assess a potential bias for patients with multiple lesions. Model 3 was included because activating *NRAS* mutations could potentially result in a similar phenotype as *BRAF*-mt, since mutations in both genes lead to activation of the mitogen-activated protein kinase (MAPK) pathway. The simple benchmark model was evaluated in a similar way as model 1, i.e., using all lesions and automatically extracted radiomics features. Model 5 was applied to compare the performance of WORC to a simple benchmark machine learning model, which uses binary logistic regression with LASSO (least absolute shrinkage and selection operator) feature selection (i.e., ElasticNet).

### 2.5. Statistics

To assess the predictive value of the individual features, the Mann–Whitney U test was performed for univariate analyses of continuous variables and Pearson’s chi-squared test was used for categorical variables. For radiomics, *p*-values were corrected for multiple testing using the Bonferroni correction according to the default in WORC. A *p*-value of <0.05 was considered to be statistically significant.

Evaluation of the radiomics models was performed using a 100× random-split cross-validation. In each iteration, the data was randomly split into 80% for training and 20% for testing in a stratified manner to guarantee a similar distribution of the classes in the training and test set as compared to the original set. Metastases from the same patients were always grouped together in either the training or test set. To eliminate the risk of overfitting, in each iteration, all model optimization was performed strictly within the training set by using a second internal 5× random-split cross-validation (see Appendix A). The final model consists of an ensemble of the 50 best workflows, i.e., combination of methods and parameters, each defined by a specific set of hyperparameters. This final model may be different in each of the 100× random-split cross-validation iterations. For each of the five models described in the experimental setup, these sets hyperparameters are included with the code [33]. Details are described in Appendix A.

The performance of all four models was described by the AUC of the receiver operating characteristic (ROC) curve, accuracy, sensitivity, specificity, negative predictive value (NPV) and positive predictive value (PPV). The positive class was defined as *BRAF*-mt. For each metric, the average over the 100 cross-validation iterations and a 95% confidence interval (CI) were reported. The 95% CIs were constructed using the corrected resampled t-test based on the results from all 100 cross-validation iterations, thereby taking into account that the samples in the cross-validation splits are not statistically independent [35]. ROC confidence bands were constructed using fixed-width bands [36].

## 3. Results

### 3.1. Study Population

In total, 103 patients were included, see Appendix A for a flowchart of patient inclusion. Characteristics of these patients and their CT scans are summarized in Table 1. The median age was 65 years (interquartile range (IQR) 52–72) and 50.5% of the patients were men. *BRAF* mutation status was either determined on the primary tumor (*N* = 20), local recurrence (*N* = 3), or metastasis (*N* = 79). In these lesions, *BRAF* p.V600E was detected in 51 patients, whereas 52 patients had *BRAF*-wt melanomas. In total, 103 CT scans were acquired from 10 different CT scanners, resulting in in the inclusion of data acquired with different acquisition protocols (Table 1). Although for all acquisition parameters the difference between *BRAF*-mt and *BRAF*-wt was not statistically significant, the difference in tube current reached almost statistical significance (*p* = 0.05).

### 3.2. Radiomics and LIDC Features and Models

In total, 169 lung metastases in 103 patients were segmented. Figure 2 illustrates randomly selected segmentations of lung metastases from patients with *BRAF*-mt and *BRAF*-wt metastatic melanoma. Median volume of segmented lung lesions was 18.3 mL (IQR: 7.3–48.6 mL). None of the radiomics or LIDC features were significantly different between *BRAF*-mt and *BRAF*-wt lung metastases, as none of the features had a *p*-value < 0.05 after Bonferroni correction. LIDC criteria scores are shown in Appendix A. Using all 169 lung metastases, the radiomics model (model 1) resulted in a mean AUC of 0.49, sensitivity of 0.61 and specificity of 0.37 (Figure 3A, Table 2). Model 2, i.e., only inclusion of the largest lesion per patient, slightly improved the performance (AUC of 0.65), whereas model 3, i.e., only inclusion of *BRAF*-wt melanoma who were also *NRAS* wild type, still had a poor performance (AUC of 0.49) (Figure 3B,C, Table 2). In addition, model 4, i.e., based on the LIDC features scored by a radiologist, resulted in an AUC of 0.46 (Figure 3D). The simple benchmark (model 5) resulted in a similar performance (AUC of 0.50).

## 4. Discussion

The results of this study show that there is no association between radiomics features of lung metastases and the *BRAF* mutation status in patients with metastatic melanoma. Our model using only the largest lesion per patient performed best with a moderate mean AUC, but still none of the features had any individual discriminative value. In addition, the performance confidence intervals (e.g., the sensitivity and specificity) still included many values below the performance of guessing. The LIDC criteria as scored by a thorax radiologist also failed to discriminate the *BRAF* mutation status in melanoma lung metastases.

Despite the remarkable success of BRAF inhibitors and immunotherapy in patients with metastatic melanoma, only a subset of patients benefits from these therapies [11,37]. Tools to select the patients most likely to benefit are of great interest and this has resulted in several radiomics studies aiming to predict tumor response. Similar to our study, previous radiomics models, either to predict therapy response or survival, had a low to moderate performance in metastatic melanoma [26,38,39]. In the largest radiomics study in melanoma thus far, 483 lesions from 80 melanoma patients were included and a greater morphological heterogeneity of lymph nodes determined by CT was associated with immunotherapy response, resulting in a moderate AUC of 0.64 [26]. However, the model performed poorly in lung and liver lesions (AUC of 0.55). Comparable to our CT-based findings, a recent study showed that radiomics features derived from ^18^F-FDG positron emission tomography (PET) to determine the *BRAF* p.V600E mutation status also had a moderate performance (AUC of 0.62). They studied 176 lesions, including 18 lung lesions from 70 patients with melanoma (35 *BRAF*-mt and 35 *BRAF*-wt) [40]. To the best of our knowledge, this PET study [40] and our CT study are the first melanoma studies aiming to predict *BRAF* p.V600E mutation status, showing that neither PET nor CT radiomics features can discriminate between patients with *BRAF*-mt and *BRAF*-wt melanomas. We therefore believe that our comprehensive study provides insight into the potential of radiomics in this area, which can guide future research [41].

The lack of discrimination between *BRAF*-mt and *BRAF*-wt melanoma could potentially be explained by activating mutations in the *NRAS* gene in *BRAF*-wt melanoma. Since *NRAS* and *BRAF* are involved in the same pathway, i.e., the MAPK pathway, activating *NRAS* and *BRAF* mutations could result in a similar phenotype. Therefore, we evaluated an additional model which only included *NRAS* wild type lesions in patients with *BRAF*-wt melanoma (model 3). In our cohort of patients with *BRAF*-wt melanoma, 22 out of 45 (49%) patients—with known *NRAS* mutation status—had a *NRAS* mutation. Exclusion of all patients with *NRAS* mutation or unknown *NRAS* mutation status resulted in an AUC of 0.54 (95% CI 0.44–0.64). Based on these findings, it is very unlikely that inclusion of *NRAS* mutant melanomas negatively impacted our results. In addition, our findings are supported by the low predictive value of PET radiomics in the same setting in which patients with *NRAS* mutations were also excluded [40].

Our study was designed for a comprehensive evaluation of the relationship between CT imaging features and the *BRAF* mutation status in melanoma lung metastases. To our knowledge this is currently the largest CT-based radiomics study on the *BRAF* mutation status in patients with metastatic melanoma and with 103 subjects even large for a radiomics study [42]. It is unlikely that treatment-related resistance mechanisms influenced the outcome, since the study population was treatment-naïve, thereby reflecting the appearance of untreated melanoma lung metastases. The investigated patient population only included melanoma patients for whom correct determination of the *BRAF* status is of utmost importance for rapid treatment stratification. The WORC radiomics method applied has been previously validated to predict mutation status of several genes in other tumor types, such as lipoma and liposarcoma [32], desmoids [43], gastrointestinal stromal tumors [44], liver cancer [29,45], prostate cancer [46] and mesenteric fibrosis [47]. In these previous studies, the radiomics models had a much better performance (mean AUCs between 0.71–0.89) and multiple features were statistically significant in univariate statistical testing. In the current study, none of the radiomics features had any discriminative value; therefore, it can be concluded that radiomics features of melanoma lung metastases are not related to the *BRAF* mutation status. WORC includes a wide variety of radiomics approaches and automatically optimizes the combination, thereby evaluating many different approaches. Moreover, a different normalization method, combining z-scoring with a logarithmic transform and a correction term to better cope with outliers and non-normally distributed features [48], yielded similar negative results (model 1: AUC of 0.49). Hence, it is unlikely that a different radiomics approach will lead to a positive result.

In addition to the radiomics analysis, a radiologist visually evaluated the lesions. Similar to radiomics results, the radiologist could not discriminate between *BRAF*-wt and *BRAF*-mt lesions by applying the LIDC criteria. Although radiomics can potentially correlate imaging features with clinical outcome even in cases where a radiologist cannot, the relation between quantitative imaging features and clinical outcome is considered stronger when clinical outcomes can be discriminated visually by a radiologist. This was not evident in the current study and it can be considered additional evidence that a CT-based radiomics signature probably does not exist for the *BRAF* mutation status in melanoma lung metastases. Although radiomics is promising in other fields of research, it is not expected that all cytogenetic changes are associated with morphological changes. Consequently, it is unlikely that every DNA alteration can be detected by radiomics.

Our study has several limitations. Firstly, the *BRAF* mutation status was often determined on other tumor tissue than the segmented lung metastases. The *BRAF* status was determined on biopsy material from a lung metastasis, which did not necessarily match the segmented lung lesion, in only 12 patients. Although the concordance rate of the *BRAF* mutation status between primary melanoma and metastases is quite high [8,49,50], a recent meta-analysis showed a pooled discrepancy rate of 13.4% between primary melanomas and metastases and a 7.3% discrepancy rate between metastatic sites [51]. Hence, tumor heterogeneity might have caused misclassification of *BRAF* mutation status, thereby negatively affecting the results. Ideally, in prospective radiomics studies, genomic and radiomics analyses are performed on the same tumor site. Secondly, the segmentation of regions of interest (ROI) was performed semi-automatically. Automatic segmentation methods may improve the consistency of the segmentations and thus affect the radiomics model. However, due to the clear distinction of lung lesions and their surroundings, it is not expected that automatic segmentation will substantially alter the results. Thirdly, the heterogeneity in the acquisition protocols may have negatively affected the performance or our radiomics model. These variations may have led to variations in the imaging features, which complicate the recognition of patterns. Using a single acquisition protocol would give an estimate of the performance unaffected by such variations. However, the variations in the acquisition protocols were small, making it unlikely this significantly affected the results of the current study. Feature selection methods based on feature test-retest reproducibility could be investigated in future work [52,53]. The difference in tube current between *BRAF*-mt and *BRAF*-wt almost reached statistical significance and could have been implicitly used by the model to distinguish these lesions. However, our results show that, despite this difference, the performance of the model was similar to guessing. Lastly, although training data were strictly separated from test data in cross-validation, we did not validate our findings on an independent, external dataset.

## 5. Conclusions

In summary, our study demonstrates that neither CT-based radiomics features, nor CT-derived LIDC features scored by a radiologist can discriminate between *BRAF* mutant and *BRAF* wild type lung metastases in patients with metastatic melanoma. Therefore, CT-based parameters cannot replace determination of *BRAF* mutation status on tumor tissue.

## Figures and Tables

**Figure 1 jpm-11-00257-f001:**
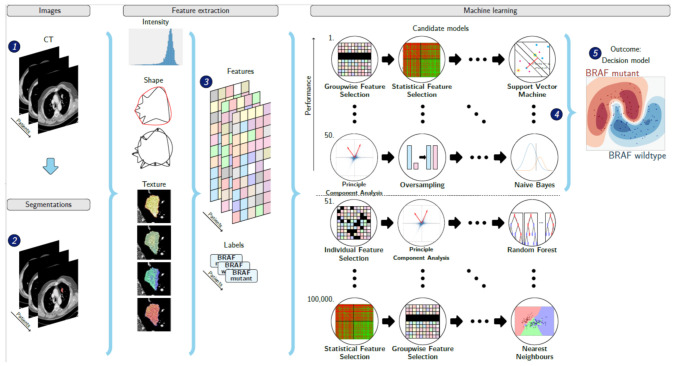
Schematic overview of the radiomics approach: adapted from Vos and Starmans et al. [32]. Inputs to the algorithm are (**1**) contrast-enhanced thoracic CT images of patients with BRAF mutated or BRAF wild type metastatic melanoma and (**2**) a segmentation of the lung metastasis. Processing steps include (**3**) feature extraction and (**5**) the creation of a machine learning decision model, using (**4**) an ensemble of the best 50 workflows from 100,000 candidate workflows, which are different combinations of the different processing and analysis steps (e.g., the classifier used).

**Figure 2 jpm-11-00257-f002:**
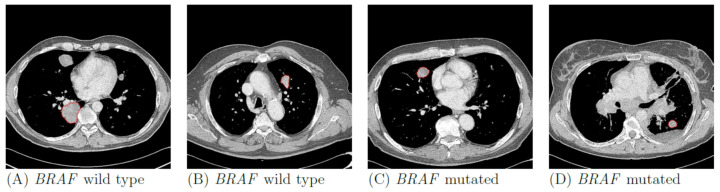
Examples of *BRAF* wild type (**A**,**B**) and *BRAF* mutant (**C**,**D**) lung metastases of four patients with metastatic melanoma. Contours of the segmentations of the selected metastases are shown in red.

**Figure 3 jpm-11-00257-f003:**
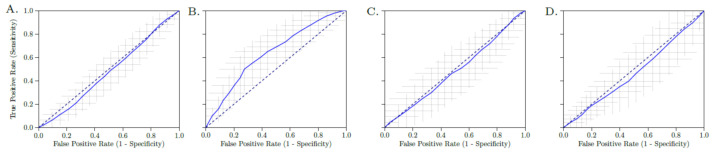
Receiver operating characteristic (ROC) curve of the radiomics model of all lesions (**A**), only the largest lesion (**B**), only *BRAF* wild type lesions with *NRAS* wild type (**C**) and LIDC features (**D**). The crosses identify the 95% confidence intervals of the 100× random-split cross-validation; the blue curve is fit through their means.

**Table 1 jpm-11-00257-t001:** Patient and imaging characteristics. Values in parentheses are percentages unless indicated otherwise.

Patient	*BRAF*-mt (*N* = 51)	*BRAF*-wt (*N* = 52)	*p*-Value
**Age (years)** ^†^	59 (50–69)	66 (57–74)	0.048
**Sex**			0.768
Male	25 (49)	27 (52)	
Female	26 (51)	25 (48)	
**Primary tumor localization**			0.027
Skin	49 (96)	42 (81)	
Mucosal	0 (0)	6 (11)	
Unknown	2 (4)	4 (8)	
**Determination of *BRAF*-mutation status**			0.851
Primary tumor	9 (18)	11 (21)	
Local recurrence	1 (2)	2 (4)	
Metastasis	40 (78)	39 (75)	
Unknown	1 (2)	0 (0)	
***NRAS* mutation status ^$^**			Not applicable
Mutant	-	22 (42)	
Wild type	-	23 (44)	
Unknown	-	7 (2)	
**Imaging**			
**Acquisition protocol**			
Slice thickness (mm) ^†,1^	1.5 (1.5, 1.5)	1.5 (1.5, 1.5)	0.23
Pixel spacing (mm) ^†^	0.68 (0.64, 0.74)	0.67 (0.61, 0.73)	0.16
Tube current (mA) ^†^	405 (278, 553)	333 (210, 490)	0.05
Peak kilovoltage ^†,1^	120 (120, 120)	120 (118, 120)	0.44
Contrast Agent			0.84
Visipaque 320	35	37	
Ultravist	1	0	
Omnipaque	1	1	
Optiray	0	1	
Unknown	14	13	
**Number of segmented lesions per patient**			0.54
One	20 (39)	17 (33)	
Two	31 (61)	35 (67)	

Values in parentheses are percentages unless stated otherwise. ^†^ Values are median (Inter quartile range). ^$^
*NRAS* and *BRAF* mutations are mutually exclusively occurring; hence, we did not test for significance between BRAF wild type versus mutant cases. ^1^ Other values than those given in the median and inter quartile range do occur.

**Table 2 jpm-11-00257-t002:** Performance of the models for *BRAF* mutation prediction based on different sets of features and lesions.

	Model 1 Radiomics All Lesions—WORC	Model 2 Radiomics Largest Lesion	Model 3 Radiomics *NRAS* Wild Type	Model 4 LIDC All Lesions	Model 5 Radiomics All Lesions—Benchmark
**AUC**	0.49 [0.38, 0.59]	0.65 [0.51, 0.79]	0.49 [0.37, 0.61]	0.46 [0.38, 0.55]	0.50 [0.42, 0.58]
**Accuracy**	0.48 [0.39, 0.57]	0.61 [0.50, 0.72]	0.65 [0.58, 0.71]	0.49 [0.42, 0.56]	0.50 [0.43, 0.57]
**Sensitivity**	0.61 [0.44, 0.77]	0.61 [0.42, 0.80]	0.94 [0.87, 1.00]	0.29 [0.11, 0.48]	0.56 [0.32, 0.80]
**Specificity**	0.37 [0.22, 0.52]	0.60 [0.38, 0.82]	0.08 [0.00, 0.17]	0.66 [0.46, 0.86]	0.44 [0.20, 0.69]
**NPV**	0.53 [0.39, 0.66]	0.61 [0.46, 0.76]	0.35 [0.00, 0.75]	0.52 [0.42, 0.61]	0.43 [0.21, 0.66]
**PPV**	0.45 [0.37, 0.53]	0.63 [0.48, 0.77]	0.67 [0.62, 0.72]	0.44 [0.30, 0.58]	0.47 [0.37, 0.56]

Abbreviations: AUC: area under the receiver operating characteristic curve; NPV: negative predictive value; PPV: positive predictive value.

## Data Availability

Imaging and clinical research data are not available at this time. Programming code is openly available on Zenodo at https://doi.org/10.5281/zenodo.4644067 (accessed on 27 January 2021).

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
