# Peer review of "The BRAF P.V600E Mutation Status of Melanoma Lung Metastases Cannot Be Discriminated on Computed Tomography by LIDC Criteria nor Radiomics Using Machine Learning"

_jpm, 2021, doi:10.3390/jpm11040257_

Round 1

Reviewer 1 Report

The aim of this study was to evaluate whether radiomics can
predict the BRAF status in a non-invasive manner.

On the basis of the conducted research  results according the BRAF mutation status in melanoma lung metastases it was shown that it cannot be predicted using radiomics features or visually scored LIDC criteria

My comment:

The cytogenetic aspects are not necessarily mirrored by morphological changes.

Reviewer 2 Report

The paper is very well structured, with a well described method, in a linear and synthetic way. The negative result of the study is however useful for analyzing strategies to overcome the highlighted limitations in future works. Some observations that may be useful to the authors in this phase of the study are also clinical limits that I believe are important to highlight.

From a technical point of view, a limitation of the study lies in the selection of features. In particular, the selection takes place within the WORC system, which selects the best method of reducing the features among the different proposals by measuring them on the best accuracy achieved. This system is useful, but it constrains accuracy to the study data itself, which can reduce reproducibility on external datasets if the results are more positive.

A system that could improve the selection of features is certainly the test-retest as already tested in the work of Rossi et al. published on Cancer research.

Another, completely missing, aspect is the consideration of clinical variables. As already known, there are clinical variables that increase the probability of finding a BRAF mutation such as those following: low tumor thickness; low mitotic rate, low Ki67 score; superficial spreading melanoma; pigmented melanoma; a lack of history of solar keratoses; a location on the trunk or extremity; a high level of self-reported childhood sun exposure; <50 years of age; and fewer freckles. (as observed in doi: 10.1038 / sj.jid.5700632). Since these are lung lesions, an information that could help the algorithm to discriminate the features of the mutant versus wildtype is the smoking status. It could alter the surrounding "healthy" lung parenchyma
and generating confusion in the algorithm (more when segmentation of a lesion is difficult as smaller lesion). The clinical limitations highlighted by the paper can somehow shed light on other possible strategies to improve the algorithm.

As mentioned there is a minimum heterogeneity between primitive and metastasis on the expression of BRAF (and not being sure that the lesion is actually BRAF is a rightly emphasized limit). we cannot exclude that this heterogeneity also exists between different lung lesions, therefore the approach that consider only one lesion (for which biological data is available) may be better.

As seen, the exclusion of NRAS patients among BRAF-wt patients slightly (too slightly) improves the algorithm. Finally, it should also be clarified whether all patients were c-KIT wt or not tested, as this can generate additional confounding features (outliers).

Furthermore, there is no harmonization of the features that could further improve the study itself compared to normalization via z-score alone.

Reviewer 3 Report

Angus et al presented very interesting data on radiomics features of lung melanoma metastases trying to find novel non invasive criteria to discriminate BRAF status.  The manuscript is well written, is methodologically rigorous and clearly states its limitations. However even if it presents negative results, in my opinion, it deserves to be continued. 

Author Response

We thank the reviewer for reviewing and the positive feedback.

Round 2

Reviewer 2 Report

The corrections define the limits of the study, making it a useful starting point for future studies.